# Anatomy in Competencies-Based Medical Education

**Erich Brenner** 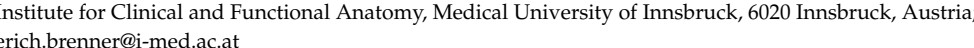

Institute for Clinical and Functional Anatomy, Medical University of Innsbruck, 6020 Innsbruck, Austria; erich.brenner@i-med.ac.at

**Abstract:** Anatomy as a basic science discipline is "vanishing" in recent competencies-based medical curricula. The fundamental requirement of these curricula to apply the knowledge from the basic disciplines in practical and clinical everyday life contributes to this disappearance. Anatomical educational objectives are in many cases not yet adapted to these changes. At the same time, the higher levels of the progress dimension in the cognitive domain and the activities associated with them certainly allow the application; even the analysis or evaluation of anatomical knowledge. However, a change in the teaching objectives to these higher levels of the progress dimension must also be accompanied by a change in the anatomical assessments. Since these forms of examinations themselves represent a practical application of anatomical knowledge, they must be carried out on suitable examination material. However, in order to protect living persons, the donated body again comes into focus.

**Keywords:** anatomical education; medical education; educational objectives; assessment; competencies

## 1. Introduction

Anatomy, and with it, histology and embryology, is one of the basic sciences of medicine. It imparts a great deal of factual and conceptual knowledge about the structure and, to some extent, the function of the human body. Anatomy is thus an essential part of the "literacy" of medical studies, enabling an inculcation of a multidimensional representation of the human body in one's mind. It shares this "fate" with other basic science subjects, such as physiology, biochemistry, or physics.

In a competency-based curriculum, the premise is that the many roles and functions involved in the doctor's work can be defined and listed clearly; and that this can be dismantled into smaller cumulative steps through which students may work at individual rates of progress through deliberate practice and formative feedback until they reach the desired level of proficiency or expertise [1].

Educational objectives from the basic sciences contribute significantly to the cognitive domain [2]. Their contribution to the psychomotor [3] and affective domains [4] can only be constructed secondarily. At the same time, anatomy still has a relatively large share in the affective domain via its ethical aspects [5,6].

Increasingly, medical studies are oriented towards competencies [7]. The CanMEDS framework [8] can be seen as an example here, "*that identifies and describes the abilities physicians require to effectively meet the health care needs of the people they serve. [ . . . ] A competent physician seamlessly integrates the competencies of all seven CanMEDS Roles*". If one looks at these seven roles, the basic sciences are only to be found "in the margins". In the central role as "Medical Expert", "*physicians draw upon an evolving body of knowledge*" [ . . . ]. In their key competence to "practice medicine within their defined scope of practice and expertise", they "*apply knowledge of the clinical and biomedical sciences relevant to their discipline*". In their role as "Collaborator", physicians should "*share knowledge and information*". In their role as "Scholar", "*physicians strive to master their domains of expertise and to share their knowledge*". By sure, anatomy contributes to the "evolving body of knowledge" of a Medical Expert; a knowledge that should be drawn upon and applied in practising medicine.

In their AMEE Guide on 'the assessment of learning outcomes for the competent and reflective physician', Shumway and Harden [9] allocate the basic sciences—and thereby anatomy—in their learning outcome 8: "*Approach practice with an understanding of basic and clinical sciences*". Doctors should understand the basic, clinical, and social sciences that underpin the practice of medicine. This ensures that they not only have the technical competence [ . . . ], but also an understanding of what they are doing, and why they are doing it. This includes an awareness of the psychosocial dimensions of medicine.

The term "apply" allows a wide range of interpretations. To "apply" can mean to say what is required or what has to be decided, to present knowledge in a written form, or to record contexts pictorially. But "applying" can also mean carrying out an activity or critically reviewing what has been done. Only these different interpretations make it possible to assess anatomical knowledge adequately in the context of a competency-based curriculum.

Whereas the words competency and competence are often used interchangeably in the literature, the term competency is used for the skill itself and competence as an attribute of the performer's ability to perform the skill [10]. The concept of competence has three main components: the first is knowledge; the second is the methodology of its application and the mastery of this methodology; and the third is a practical skill [11]. In the past, the emphasis was on knowledge; nowadays, the emphasis is more on the development of practical components. An extreme case of this approach is that competences can also be developed without a direct reliance on knowledge [11].

## 2. Educational Objectives in Anatomy

*"Tomorrow's Doctors" and the Educational Objectives of the Anatomical Society*

The General Medical Council's "Tomorrow's doctors" defined that "the graduate will be able to apply to medical practice biomedical scientific principles, method and knowledge relating to: anatomy, biochemistry, cell biology, genetics, immunology, microbiology, molecular biology, nutrition, pathology, pharmacology and physiology." [12]. Therefore, "the graduate will be able to [ . . . ] explain normal human structure and functions". In the 2020 revision, the General Medical Council expanded this teaching objective somewhat, particularly to include the aspect of development over time: "Newly qualified doctors [ . . . ] must be able to explain how normal human structure and function and physiological processes applies, including at the extremes of age, in children and young people and during pregnancy and childbirth" [13]. There is no further definition of the necessary and expected anatomical knowledge.

In order to fill this very general educational objective with content, the Anatomical Society developed a core regional anatomy syllabus for undergraduate medical education; this comprises 156 educational objectives [14]. Each of these educational objectives includes various anatomical entities, which are ultimately introduced with a verb—mostly. In the analysis, only eight different verbs are found, which are used 241 times in the 156 educational objectives (Table 1). These verbs are basically part of the cognitive domain, as the perennial question for the development of this core syllabus was: "What do I need to know?".

**Table 1.** Verbs in the anatomical objectives of the Anatomical Society [14].

| Verb | Counts |
| --- | --- |
| describe | 127 |
| explain | 54 |
| demonstrate | 25 |
| identify | 13 |
| name | 8 |
| interpret | 6 |
| define | 4 |
| be able | 4 |

### 3. Evolution of Educational Objectives

In order for the educational objectives in a competencies-based curriculum to be used on the essential aspect of "apply", certain developments in terms of the criteria of "application" are needed. It is certainly not enough to add an "apply in a clinical context", especially when programme directors, vice-deans, and vice-rectors responsible for teaching, or rectors are not anatomists.

The General Medical Council defines three levels of competence [15]:

- Safe to practice in simulation;
- Safe to practice under direct supervision;
- Safe to practice under indirect supervision.

However, these competence levels are not so easily applicable to the cognitive domain. Bloom's basic taxonomy refers exclusively to the cognitive domain [2]. In doing so, the authors introduced six levels:

(a) Knowledge;
(b) Understanding;
(c) Application;
(d) Analysis;
(e) Synthesis;
(f) Evaluation.

In a simplified form, Miller also reproduced this system with his well-known Miller's Pyramid, which in turn already introduced active verbs ("knows", "knows how", "shows how", and "does") [16]. This transformation into active activity words, in turn, facilitates the development of teaching objectives enormously. Finally, Lorin W. Anderson, a former student of Bloom's, also referred to this in her revision of the taxonomy [17]. With a further "refinement", Krathwohl [18], co-author of both Bloom's and Anderson's work, took an important step by splitting the cognitive domain into two dimensions: a dimension of cognitive processes and a dimension of knowledge. However, since the term "cognitive" was (and is) used twice here, i.e., for the domain as a whole and for one of its dimensions, Brenner and Pierer [19] suggest changing the designation of the dimensions to a "content dimension" and a "progress dimension". The content dimension of the cognitive domain begins with pure factual knowledge, and extends through conceptual knowledge and action knowledge to metacognitive knowledge. The progress dimension begins with simple repetition; and reaches the highest level of evaluation and synthesis via understanding, application, and analysis. Samples of educational objectives with differences in the content and progress dimensions are given in Table 2.

**Table 2.** Samples of educational objectives.

| Content Dimension | Progress Dimension | Educational Objective |
|---|---|---|
| Factual | Repetition | The student reports the sequence of the sections of the small intestine. |
| Factual | Evaluation | The student critically compares different findings on the attachment behaviour of the latissimus dorsi muscle. |
| Conceptual | Understanding | The student explains the basic similarity in the structure of the skeleton of the upper and lower extremities. |

For the psychomotor domain, this concept of two dimensions was actually introduced earlier [3,20]. Here, the content dimension includes manual skills, perceptual skills, and (psycho-) social skills (esp. communicative skills). The progress dimension begins with perception; i.e., the process of perceiving things, qualities, and/or relationships. This is followed by the readiness to deal with these things, qualities, and/or relationships. Next comes the guided response; then, the mechanism or habitual response. While these stages

are still relatively simple, the complex overt response is a stage where uncertainties can be resolved. The final stage is automatic performance. Guilbert proposes a simplification into only three stages: imitation, control, and automaticity [21].

Although the affective domain is seen by many teachers as very complex and difficult, an analysis of its content and progress dimensions is relatively easy [4,17]. The content dimension describes whether and how people react appropriately to emotional events. The content dimension thus includes above all attitudes, (general) perceptions, and behaviours. These contents are influenced by many other factors, such as families, religions, and societies. Knowledge of these contents of the affective domain makes it possible for teachers to modify it through targeted interventions. The progress dimension includes receptivity, the (adequate) response to concrete impetus, and internalisation.

When developing educational objectives, the combination of an educational content (what?) with an activity (does?), a learner (who?), and a criterion (how well?) is necessary. The teaching content already exists, the activity is presented as verb, and the learner is defined as graduate of medical education; very often, however, the necessary criterion is missing. Once the teachers are clear about the teaching content, the next step is to define what the learners should do with this teaching content, how well. The taxonomy is of great help here, as it can be used to reflect the essential progress in a standardised form (progress dimension). This then leads quite casually to the last step in the development of the teaching objectives, in respect of the criterion, "How well"?

However, the criterion, "How well?" is difficult to define, especially for the cognitive domain. The competence levels already mentioned above can only be applied to a very limited extent or not at all. Thus, the question arises as to how the higher levels of the progress dimension of the cognitive domain can be achieved. According to the CanMEDS role as a medical expert, the acquired knowledge of the basic sciences should be "applied".

The key to resolving this dilemma is to clarify how anatomical knowledge can actually be applied; perhaps even used to analyse practical clinical situations. Table 3 lists possible related activities (verbs) for each stage of the progression dimension. The practical or clinical application, and analysis of anatomical knowledge may relate to surface anatomy, physical examination, endoscopic anatomy, surgical approaches, or the entire spectrum of imaging techniques [22]. This means that corresponding competence levels can then also be implemented.

**Table 3.** Related activities for each level of the progress dimension.

| Progress Dimension | Related Activities (Verbs) |
|---|---|
| Knowledge | imply, indicate, state, recite, write down, enumerate, express, perform, state, name, report, describe, depict, tell, name, write, sketch, draw |
| Understanding | derive, determine, present, depict, define, demonstrate, interpret, explain, elucidate, formulate, highlight, identify, present, draw conclusions, translate, transfer, summarise |
| Application | apply, search, fill out, edit, calculate, print, perform, set up, enter, elaborate, determine, calculate, create, format, design, find out, produce, configure, delete, solve, use, plan, save, store |
| Analysis | analyse, determine, classify, divide, extract, contrast, highlight, isolate, sort, test, distinguish, examine, compare |
| Evaluation | select, evaluate, justify, judge, assess, decide, evaluate, critically compare, examine, take a stand, judge |
| Synthesis | derive, relate to, design, develop, relate, conceive, coordinate, order, tabulate, connect, associate, compile. |

*Note.* Underlined verbs have been identified from the Anatomical Society's anatomical educational objectives in Table 1.

## 4. Assessments with the Evolved Educational Objectives

The assessment of an understanding of anatomy as a basic science was and still is heavily concentrated in the cognitive domain [9]. The classic approach to the assessment

of anatomy is through written tests. Portfolios and logbooks might also be helpful in that they require students to reflect on the relationship of what they know to the application of what they do in a care setting.

When assessing medical knowledge, distinguishing between the acquisition of knowledge and its application is critical. When assessing the acquisition of medical knowledge, the outcome is to document clinically applicable knowledge of the basic and clinical sciences—such as anatomy—that underlie the practice of medicine. When assessing the application of medical knowledge, the goal is to assess the ability to apply that knowledge to clinical problem-solving and clinical reasoning [23].

Once educational objectives have been defined that answer the question, "who does what and how well", the (over-)examination can be defined without further ado. In the process, it often turns out that the originally planned forms of examination do not correspond to the teaching objectives by far. A written multiple-choice examination will be largely unsuitable for an educational objective in the psychomotor domain; rather, the teacher will not be able to avoid a demonstrative form of assessment for performance evaluation. If, for example, a manual skill at the level of automatism is to be the educational objective, the corresponding performance assessment must also take automatism into account; i.e., multiple repetitions of the skill, perhaps even under changing environmental parameters (settings).

If we now take into account the higher levels of the progress dimension, and their application to the practical or clinical application of anatomical knowledge, the forms of assessment must also change accordingly. Generally, simple written examination forms such as a multiple-choice test are assigned to the basal levels of knowledge reproduction or comprehension. However, well-developed MC questions can be quite useful. Spot examinations (so-called "spotters") also belong to the widespread forms of assessment [24,25]. However, these spotters are fundamentally based on the activity of "identifying" an anatomical entity; and are thus at the level of understanding. Often the-correct-identification is followed by questions in a functional or clinical context; in sum, called an objective structured practical examination (OSPE) [26,27]. Nevertheless, the initial "identification" task of a spotter, and the functional or clinical questions may also be assessed independently [27].

In general, the assessment of medical knowledge application is a more complex process with variable validity and reliability. The implementation of such assessments should be carefully planned, and address where and how the assessment will be used by the program [23].

But how can surface anatomy, physical examination, endoscopic anatomy, surgical approaches, or the entire spectrum of imaging techniques become part of a challenging competency-based examination? For surface anatomy, for example, it would be possible not to "present" the students with pre-marked specimens, but to ask them to place the corresponding marks themselves. For the physical examination, the examination task could be the palpation of a specific anatomical structure. Surgical approaches could be assessed by the candidate actively demonstrating this access route. Norcini and Burch [28] present several tools for assessing such competencies, mainly by direct observation. By sure, these assessment tools must be adopted for anatomical purposes. For all these forms of assessment in anatomy, the occasionally disdained body donors are a good choice; since here, the corresponding activities can be carried out without endangering living persons—even several times.

## 5. Conclusions

The basic science discipline of anatomy can therefore make a significant contribution to the required basic knowledge and, above all, its application; especially today in competency-based curricula. Traditional anatomical educational objectives for anatomy target mainly the factual knowledge in the cognitive domain, as also the sample analysis of the Anatomical Society's core regional anatomy syllabus for undergraduate medical education [14]. This factual knowledge is by sure a core competency; however, it is hidden

within other competencies. Nevertheless, the subject of anatomy has thus become virtually invisible within the framework of these educational objectives. Revising the educational objectives towards more application, analysis, and synthesis could broaden the anchoring of anatomy to further competencies. However, this means that the already existing educational objectives must be refined in order to be able to meet the new requirements as well. Parallel to this, the forms of the examination must also be adapted.

**Funding:** This research received no external funding.

**Institutional Review Board Statement:** Not applicable.

**Informed Consent Statement:** Not applicable.

**Data Availability Statement:** Not applicable.

**Conflicts of Interest:** The author declares no conflict of interest.

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
