# Peer review of "Anatomy in Competencies-Based Medical Education"

_education, doi:10.3390/educsci12090610_

Round 1
Reviewer 1 Report
This article is generally well-written. The authors propose ‘evolved ways to apply anatomy’ and ways to assess them.
Besides providing a list of verbs to be used parallel with the few commonly used for learning outcomes, it is hard to see the main argument of the paper. Abstract is not aligned with the text, neither are the conclusions. The abstract is concluded “..donated body again comes to focus.”, but cadavers or dissections are not discussed in the text.
Lines 65-67 may present the statement/hypothesis; competency maybe achieved without knowledge. Authors argue for such a case (ref.11.), which is unfortunately published in Russian (I have no competency nor knowledge of this language). Based on the ref 11 abstract (which is provided in English) it is impossible to understand how the study was conducted, size of the cohorts or anything else relevant. No other evidence is provided competence without knowledge is provided.
There are also structural problems, such as the Tables are not referred in text at all.
Author Response
Thank you very much for your comments!
Lines 65-67: competency without knowledge: This is the only reference I found on this topic. I'm sorry that the reference is written in Russian language. (Probably a translation software could help?).
Cadavers or dissections are addressed in the text at lines 216/217.
The references to the tables were corrupted by the system; corrected.
Reviewer 2 Report
Dear authors,
The study’s aim is unclear and lacks methodology; in addition, the study some sentences are highly similar to the web and journals; please check with the attachment.
Thank you.

Author Response
Thank you very much for your comments.
This is not a research study, but a review article in the form of a perspective. Each of the references has been carefully checked and properly cited. Therefore similarities do appear ...
Reviewer 3 Report
Thank you for the opportunity to review this interesting review-article on anatomy in competencies-based medical education.
I have just some minor comments.
#1 There is some inconsistence of using large and lower case. For example: line 39 and 44 "Medical Expert", line 42 "Collaborator", line 43 "Scholar"
#2 In line 36, 39, 52, 74, 77: please explain or correct "[...]"
#3 line 62 typo: "7The concept..."
#4 line 86, 125, 161: please correct "(Error! Reference source not found.)"
#5 line 130: "(psycho-)social" insert space > "(psycho-) social"
#6 references: please check all references and adhere to the style (e.g., bold years)
Author Response
Thank you very much for your efforts!
@#1: These capitalisations are citations from the CanMEDS documents; therefore capitalisation has been taken over.
@#2: [...] is used as truncation; it would be too extensive to cite the complete texts from the CanMEDs roles here.
@#3: corrected.
@#4: corrected (these were the references to the tables).
@#5: corrected.
@#6: MDPI-reference style for EndNote - as provided by the publisher - was used. Formatting of the references should therefore meet the publisher's guidelines.
Reviewer 4 Report
Firstly, thank you for opportunity to review very interested article. I don't feel qualified to judge about the English language and style due to not native language.
1. The title reflect the main subject about anatomy in medical education, title was clear and easy to understand.
2. The abstract summarize and reflect the work described in the manuscript.
3. The key words reflect the focus of the manuscript.
4. The manuscript adequately describe the background, present status, and significance of the study. The authors explain anatomy in many aspects and education policy in the first part of introduction. However, I suggested the authors to demonstrate anatomy in many countries that's different education process.
- in line 86, 125, 161 Error! Reference source not found
- in line 76 ,, to "
- in line 74, 77 […]
5. The manuscript interpret the findings adequately and appropriately, highlighting the key points concisely, clearly, and logically.
6. Tables sufficient, good quality and appropriately illustrative of the paper contents.
7. The manuscript cite appropriately the latest, important, and authoritative references in the introduction and discussion sections. However, some of references were incorrect style for this journal.
Author Response
Thank you very much for your efforts!
line 76: corrected
[...]: standard in-text truncation, no correction needed.
MDPI-provided EndNote-Style was used, therefore the references should be formatted correctly.
Round 2
Reviewer 2 Report
The study presented second hands references and lacked research design, questions, hypotheses, and methods; all of them are unclearly stated. Thus I am not sure that could contribute to readers.
Thank you.